# Aluminosilicates as a Double-Edged Sword: Adsorption of Aflatoxin B_1_ and Sequestration of Essential Trace Minerals in an In Vitro Gastrointestinal Poultry Model

**DOI:** 10.3390/toxins15090519

**Published:** 2023-08-24

**Authors:** Sara Paola Hernández-Martínez, Armando Delgado-Cedeño, Yareellys Ramos-Zayas, Moisés Armides Franco-Molina, Gerardo Méndez-Zamora, Alicia Guadalupe Marroquín-Cardona, Jorge R. Kawas

**Affiliations:** 1Facultad de Agronomía, Universidad Autónoma de Nuevo León, Escobedo 66050, Nuevo León, Mexico; sara.hernandezma@uanl.edu.mx (S.P.H.-M.); gerardo.mendezzm@uanl.edu.mx (G.M.-Z.); 2MNA de México, Juárez 67250, Nuevo León, Mexico; adelgadocn@uanl.edu.mx (A.D.-C.); yramosz@uanl.edu.mx (Y.R.-Z.); 3Facultad de Medicina Veterinaria y Zootecnia, Universidad Autónoma de Nuevo León, Escobedo 66050, Nuevo León, Mexico; 4Facultad de Biología, Universidad Autónoma de Nuevo León, San Nicolás de los Garza 66455, Nuevo León, Mexico; moyfranco@gmail.com

**Keywords:** montmorillonite, beidellite, clinoptilolite, aflatoxin B_1_, trace inorganic nutrients, in vitro poultry gastrointestinal model

## Abstract

Aflatoxins can cause intoxication and poisoning in animals and humans. Among these molecules, aflatoxin B_1_ (AFB_1_) is the most dangerous because of its carcinogenic and mutagenic properties. To mitigate these effects, clay adsorbents are commonly included in the diet of animals to adsorb the carcinogens and prevent their absorption in the gastrointestinal tract. In this study, four clays, three smectites (C-1, C-2, and C-3), and one zeolite (C-4), were compared as adsorbents of AFB_1_ and trace inorganic nutrients using an in vitro gastrointestinal model for poultry. Characterization of the clays using Fourier transform infrared spectroscopy revealed characteristic bands of smectites in C-1, C-2, and C-3 (stretching vibrations of Si-O, Al-O-Si, and Si-O-Si). The C-4 presented bands related to the bending vibration of structural units (Si-O-Si and Al-O-Si). X-ray diffraction analysis showed that C-1 is a montmorillonite, C-2 is a beidellite, C-3 is a beidellite-Ca-montmorillonite, and C-4 is a clinoptilolite. The elemental compositions of the clays showed alumina, silica, iron, calcium, and sodium contents. The cation exchange capacity was higher in C-3 clay (60.2 cmol(+)/kg) in contrast with the other clays. The AFB_1_ adsorption of C-1 was the highest (98%; *p* ˂ 0.001), followed by C-2 (94%). However, all the clays also sequestered trace inorganic nutrients (Fe, Mn, Zn, and Se). Both smectites, montmorillonite and beidellite, were the most suitable for use as adsorbents of AFB_1_.

## 1. Introduction

Aflatoxins (AFs) are toxic compounds that can contaminate a wide variety of cereals, such as corn, as well as ground nuts, pistachio, almond, walnut, rice, dried fruits, meat, and milk-based products. The consumption of these products can affect the health of humans and animals [1,2,3]. AFs are produced by fungi such as *Aspergillus flavus*, *Aspergillus parasiticus*, and *Aspergillus nomius*, which usually grow in the warm conditions of tropical and subtropical regions worldwide [4]. More than 20 types of AFs are present in nature, with the most common ones being AFB_1_, AFB_2_, AFG_1_, and AFG_2_; AFM_1_ and AFM_2_ are the secondary hydroxylated metabolites of AFB_1_ and AFB_2_ [5]. The effects of AFs toxicity include nausea, vomiting, abdominal pain, convulsions, and other signs of acute liver injury, which may trigger hepatocellular carcinoma [6]. AFs are metabolized in the liver by microsomal cytochrome enzymes (CYP450s), which form AFB_1_-exo-8,9-epoxide [7]. The AFB_1_-8,9-epoxide metabolite forms adducts with amino acids and DNA that can cause gene mutations and result in cancer [8]. These adducts are used as biomarkers because they are detectable in the blood (AFB_1_-lysine adduct), urine (AFB_1_-N7-guanine adduct), milk (AFM_1_), and tissue samples (AFB_1_) [9,10]. In 1993, the International Agency for Research on Cancer classified AFs as Group 1 carcinogens because of their toxic, carcinogenic, mutagenic, teratogenic, and immunotoxic nature [11,12].

To avoid or reduce the risks posed by AFs to the health of animals and humans, strict regulations have been implemented to prevent AFs contamination in foods and animal feed. The US Food and Drug Administration (FDA) established a level of 20 μg/kg for AFs in all foods [13], whereas the European Union established levels of 2–4 μg/kg for human foods and 0.5 μg for milk products [2].

Chemically, AFs consist of a bifuran ring fused to a coumarin nucleus with a pentanone ring (AFB and AFM) or six-membered lactone ring (AFG) [14,15]. These molecules induce fluorescence, which can be detected (AFB_1_ and AFB_2_ = blue, AFG_1_ = green, and AFG_2_ = green blue). The principal methodologies used to detect AFs are high-performance liquid chromatography (HPLC), mass spectroscopy, enzyme-linked immunosorbent assay (ELISA), radioimmunoassays, chemiluminescence immunoassays, and immunosensors [16,17].

The negative effects observed with AFs depend on the species, dose, and duration of exposure. In poultry, susceptibility varies with the species (turkey > ducks > quail > broilers > hens) [18], affecting liver function and altering immunologic, digestive, hematopoietic, and reproductive functions. This leads to a decrease in growth, feed efficiency, and egg production, resulting in economic losses for the poultry industry [19].

The AFs content and contamination of food samples depend on the harvest time, temperature, and humidity, and are increased by improper drying and poor storage [20,21]. To prevent contamination and exposure, several physical (sorting, cleaning, screening, and high-temperature), chemical (alkaline or acidic conditions), and biological (bacteria cells, yeasts, and clay minerals) methods have been developed [22,23].

Several strategies can be used to prevent the effects of AFs, including natural antioxidants such as curcumin, a molecule extracted from *Curcuma longa* (turmeric). This molecule reduces oxidative stress, cytotoxicity, and DNA damage [24]. Wang et al. (2022) observed that curcumin reduces the effects of AFB_1_ on the kidneys of mice, inhibits apoptosis by the Bax/Bcl-2–Cyt-c signaling cascade, and modulates the Keap1–Nrf2 signal pathway to enhance renal antioxidant capacity [25]. Another molecule with the same effect is resveratrol, a polyphenolic phytoalexin present in grapes that reduces the reactive oxygen species (ROS) produced by AFB_1_ [26,27]. Another strategy is the utilization of atoxigenic *A. flavus* strains (non-aflatoxin) to limit aflatoxin contamination [28]. The atoxigenic strain decreased the production of AFs through competition for nutrients and regulation of aflatoxin biosynthesis [29].

In animal nutrition, clay minerals are used as adsorbents of AFs, counteracting their toxic effects. These minerals bind to toxins, phytotoxins, enterotoxins, bacteria, and viruses to prevent their absorption in the gastrointestinal tract [24]. The binding capacity of clays is specific and depends on their structure, even if they belong to the same family [30]. Clay minerals are a diverse group of aluminosilicates composed of tetrahedral (Si^4+^, Al^3+^, and Fe^3+^) and octahedral (Al^3+^, Fe^3+^, Fe^2+^, and Mg^2+^) sheets, with both containing oxygen and hydroxyl groups [31]. The two main groups of aluminosilicates are phyllosilicates and tectosilicates. Phyllosilicates are characterized by a 1:1 (one tetrahedral and one octahedral sheet), 2:1 (one octahedral sheet between two tetrahedral sheets), or 2:1:1 (basic 2:1 structure with an interlayer containing Mg^2+^, Fe^2+^, or Al^3+^ ions) layered structure [31,32].

Smectites belong to a subgroup of phyllosilicates that includes montmorillonite and beidellite. They are characterized by a charged octahedral sheet, which originates from the substitution of Mg^2+^ for Al^3+^ in montmorillonite and Al^3+^ for Si^4+^ in the tetrahedral sheet of beidellite [33,34]. These changes allow the formation of hydrogen bonds between inorganic and organic polar molecules [35,36].

Tectosilicates exhibit a tetrahedral structure with large pores and channels and consist of a central silica atom surrounded by four oxygen atoms bound to the other tetrahedral structures, resulting in a three-dimensional network [37]. Within this group, clinoptilolite is one of the most abundant natural zeolites, possessing a two- or three-dimensional open framework structure formed by a network of (AlO_4_)^4−^ and (SiO_4_)^5−^ ions linked by oxygen atoms [38,39]. Its porous, negatively charged structure can be occupied by cations (Ca^2+^, K^+^, Na^+^), the OH group, and water molecules, which undergo a cation exchange with the other molecules in the surroundings [40]. Clinoptilolite is commonly used for water treatment, catalysis, and nuclear waste management, in agriculture and biochemical applications, and as an animal feed additive [41]. 

Previous in vivo studies reported that montmorillonite and clinoptilolite added to broiler diets increased organ weights and decreased the fat content in breast meat [42], improved the ileal digestibility of protein [43], increased the daily weight gain, decreased the serum levels of total protein, and reduced the amount of AFB_1_ residues in the liver [44]. 

This study aimed to evaluate the chemical and mineral compositions of three smectites and one zeolite and determine how their composition affects the adsorption of AFB_1_ and trace inorganic nutrients. Although several articles have addressed in depth the mechanisms of how the clay minerals interact with the aflatoxins, no research has focused on the trace inorganic nutrients adsorption or possible competition with aflatoxins for the adsorption sites on clay minerals. To elucidate the adsorption mechanisms of the clays, attenuated total reflection (ATR) Fourier transform infrared (FTIR) spectroscopy, X-ray diffraction (XRD) analysis, X-ray fluorescence (XRF), scanning electron microscopy (SEM), and cation exchange capacity (CEC) measurements were performed. The AFB_1_ and trace inorganic nutrient adsorption of the clays were evaluated using an in vitro poultry gastrointestinal model.

## 2. Results

### 2.1. Characterization of Clay

FTIR spectra were used to determine the interaction of AFB_1_ and trace inorganic nutrients with the main functional groups of the clays. Figure 1 shows the spectra of C-1, C-2, C-3, and C-4 clays before and after AFB_1_ and trace inorganic nutrient adsorption. C-1, C-2, and C-3 clays showed the characteristic bands of smectites, with the vibrational bands corresponding to the H-O-H bending of water between 3600 cm^−1^ and 1600 cm^−1^, the strong band corresponding to the stretching vibration of Si-O at 1000 cm^−1^, and bands related to the bending vibration of Al-O-Si (515 cm^−1^) and Si-O-Si (460 cm^−1^). C-4 clay showed bands at 3717 cm^−1^ and 1629 cm^−1^ related to vibration of OH groups in water and bands at 1021 cm^−1^ and 603 cm^−1^ corresponding to bending vibration of structural units [Si-O (Si, Al)]. The interaction of smectites and clinoptilolite with AFB_1_ and trace inorganic nutrients was with the main functional groups (Si-O, Al-O, and [Si-O (Si, Al)]).

The elemental compositions of the clays are listed in Table 1. All the samples exhibited high contents of Al_2_O_3_ (50–73%) and SiO_2_ (9–14%). The other minerals included MgO in C-1 clay (5.02%), Fe_2_O_3_ (5.47%), CaO (9.48%) in C-3 clay, and K_2_O (4.99%) in C-4 clay.

The mineralogical compositions are listed in Table 2. Using XRD analysis (Figure 2), C-1 clay could be characterized as Ca-montmorillonite (61.0%), with diffraction peaks at 2θ = 5.8, 19.8, 21.8, 26.5, 35.8, and 61.8, which indicated a dioctahedral structure (Figure 2A), while C-2 clay and C-3 clay contained 55.0% and 36.5% beidellite, respectively. In the case of C-2 clay, beidellite was identified as the main phase based on the diffraction peaks at 2θ = 7.1, 14.3, 19.7, and 28.5 (Figure 2B), while C-3 clay contained beidellite-Ca-montmorillonite, with diffraction peaks at 2θ = 7.0, 19.9, 26.6, 27.9, 29.0, and 61.9 (Figure 2C). Finally, C-4 clay was composed of 80.0% clinoptilolite, with diffraction peaks at 2θ = 9.8, 11.1, 22.3, 26.0, 28.0, and 31.9 (Figure 2D). After the adsorption of AFB_1_ and trace inorganic nutrients, C-1, C-2, and C-4 clays presented a lower intensity of the peaks corresponding to Ca-montmorillonite, beidellite, and clinoptilolite, respectively, and C-3 clay showed a lower and a shift in the intensity of the peaks.

The surface structures and morphologies of the adsorbents were determined using SEM (Figure 3). C-1 clay consisted of porous individual particles 3–10 µm in size and aggregates of these particles 30–60 µm in size, both of which had irregular morphologies. C-2 clay consisted of dispersed particles (2–15 µm) and their aggregates (30–50 µm), while C-3 clay consisted of irregular porous particles (2–80 µm), and C-4 clay consisted of aggregates (10–60 µm) of spherical particles (1–5 µm).

### 2.2. Determination of CEC

The CEC values of the various clays are presented in Table 3. C-3 clay showed the highest CEC value (60.18 cmol(+)/kg), followed by C-1 clay (57.44 cmol(+)/kg), C-2 clay (54.72 cmol(+)/kg), and C-4 clay (38.29 cmol(+)/kg).

### 2.3. Quantification of AFB_1_ in an In Vitro Poultry Gastrointestinal Model Using UPLC

The UPLC chromatograms showed the presence of AFB_1_ in the in vitro model for a retention time of 4 min. The amount of AFB_1_ in the supernatant decreased after adsorption by C-1, C-2, C-3, and C-4 clays. Based on these data, it was concluded that the AFB_1_ adsorption in the in vitro model was the highest (*p* = 0.001) in the case of C-1 clay (0.199 ng/kg ± 0.00012; 99.5%), followed by C-2 clay (0.184 ng/kg ± 0.00013; 92.0%), C-3 clay (0.176 ng/kg ± 0.00016; 88.4%), and C-4 clay (0.161 ng/kg ± 0.0004; 81.0%) (Table 4; Figure 4).

### 2.4. Trace Inorganic Nutrient Adsorption in an In Vitro Poultry Gastrointestinal Model

C-3 clay exhibited the highest adsorption (*p* = 0.0001) rate for Fe (95.5% ± 3.0), Se (94.5% ± 1.5), Zn (72.0% ± 2.8), and Mn (52.5% ± 3.5). The adsorption percentages for the other clays were as follows: C-2 (Fe = 72.0% ± 1.4, Se = 71.0% ± 1, Mn = 17.0% ± 1.2, and Zn 10.0% ± 0.8); C-1 (Fe = 66.0% ± 0.6, Se = 31.0% ± 1, Mn = 12.0% ± 0.4, and Zn = 6.0% ± 0.1); and C-4 (Fe = 58.5% ± 0.1, Se = 58.0% ± 0.7, and Zn = 38% ± 0.3, and Mn = 35.0% ± 0.8) (Figure 5).

## 3. Discussion

### 3.1. Characterization of Clay

The bands observed in the FTIR spectra of C-1, C-2, and C-3 clays (525 cm^−1^ and 470 cm^−1^) corresponding to Al-O-Si and Si-O-Si, presented in smectites [45,46].

The bands observed in C-4 clay correspond to the stretching and bending vibrations characteristic of Si–O–(Si, Al) groups, located between 1100 cm^−1^ and 700 cm^−1^ [47,48]. The FTIR spectra of clays after adsorption of AFB_1_ and trace inorganic nutrients showed a hypsochromic shift in the adsorption bands corresponding to their main functional groups, indicating an interaction between them.

The interaction between AFB_1_ and montmorillonite occurred within the O-Si-O of the tetrahedral sheets by chemisorption with the formation of double hydrogen bonds [49,50]. In the case of clinoptilolite clay, the adsorption of AFB_1_ is influenced by functional groups and inorganic cations from the external surface [51], and its higher sorption capacity was explained by the presence of a fairly ordered system of macro-, meso-, and microchannels [52].

Aluminosilicates are composed of silica, alumina, Mg, Fe, K, Na, and Ca [53]. In smectites a silica content of 65–68% [54] and an alumina content of 13.2% and 24.8% have been reported [55,56]. These minerals were the main components of the C-1, C-2, and C-3 clays. However, the Mg and Ca contents were highest in C-1 clay and C-3 clay, and the Fe content was higher in C-3 clay, in contrast to other reports [54]. C-4 clay had the highest contents of Fe, Ca, and Mg but lower contents of K, Na, and silica, in contrast with what was reported by Ruíz-Baltazar and Pérez (2015) [57], where clinoptilolite presented a content of Fe (0.66%), Ca (0.75%), Mg (0.53%), K (6.42%), Na (1.4%), and silica (77.4%).

The primary mineralogical component of C-1 clay was Ca-montmorillonite (61%), followed by cristobalite (21%). These results are similar to those observed by Damian et al. (2021) [58], where the main component of smectites was montmorillonite (60–95%). Adikary et al. (2015) [59] reported that the composition of smectites also included cristobalite (3–30%) and other impurities such as quartz and kaolinite.

The characteristic XRD peaks of C-1 clay correspond to Ca-montmorillonite [60,61]. Another component of smectites is beidellite [62]; this mineral was observed in both C-2 clay (55.0%) and C-3 clay (36.5%), with the corresponding XRD peaks. Santos et al. (2018) [63] observed reflexions of beidellite at 2θ = 7.1 and 14.2°.

In addition, the XRD spectrum of C-3 clay showed diffraction peaks corresponding to beidellite (36.5%), Ca-montmorillonite (15.0%), and calcite (12%) [63,64]. Clinoptilolite (80%) was the main component of C-4 clay, whose XRD pattern contained diffraction peaks characteristic of this mineral [65]. The peaks of the XRD pattern are vital to identifying the properties of materials. The changes observed in the XRD pattern of clays after adsorption confirm the interaction with the main functional groups observed in FTIR. These changes are due to substitution of elements, structural transformation, size, and lattice strain [66]. Akpomie and Dawodu (2016) [67] reported changes in the peaks of montmorillonite treated with acid medium, indicating an alteration in the alumina content. Similarly, effects were observed in beidellite intercalated with chromium [63] and in clinoptilolite intercalated with cadmium [68].

SEM was used to identify the microstructures (size and shape) of the adsorbents [69]. C-1 clay consisted of porous individual particles and aggregates [70]. However, De León et al. (2015) [71] and Yin et al. (2016) [72] reported that montmorillonite consists of plates and flat particles. C-2 clay consisted of dispersed particles and aggregates, while C-3 clay consisted of porous irregular particles, in contrast to the results reported by Yanagisawa et al. (1995) [73] and Kloprogge (1993) [74], who observed small flakes. In the case of C-4 clay, the morphology consisted of spherical particles, in contrast to the results reported by Belousov et al. (2019) [70], who observed a leaf-like structure.

### 3.2. CEC Values

The CEC is related to the substitution of ions in the tetrahedral or octahedral sheets of aluminosilicates [75]. The CEC value of C-1 clay (57.4 cmol(+)/kg) was similar to that reported by Calabria et al. (2013) [76], who observed values between 46.5 and 73.15 cmol(+)/kg. However, Choo and Bai (2016) [77] and Rihayat et al. (2018) [78] reported values greater than 75 cmol(+)/kg. In this study, C-4 clay exhibited a CEC value of 38.3 cmol(+)/kg, however, Wiyantoko and Rahmah (2017) [79] reported a CEC of 41.3 cmol(+)/kg for natural zeolite, and this increased when zeolite was active by chemical (90.9 cmol(+)/kg) and physical (181.9 cmol(+)/kg) methods. Similar results were observed by Znak et al. (2021) [52], where a sample of clinoptilolite presented a CEC value of 105.9 cmol(+)/kg, and when modified with AgNO_3_, it increased to 176.9 cmol(+)/kg.

As previously reported, the CEC value is affected by the pH, concentration of the ionic species, and presence of impurities [80]. Higher CEC values are indicative of higher concentrations of Ca^2+^, Mg^+^, Na^+^, and K^+^ ions [81].

### 3.3. Quantification of AFB_1_ Adsorption Capacity

The AFs adsorption capacity of smectites and zeolites depends on the number of active sites located on the interlayer, basal planes, surface, pores, and edges of the particles, as well as the exchangeable cations [51]. The results showed that C-1 clay exhibited the highest adsorption capacity for AFB_1_, followed by C-2, C-3, and C-4 clays. These results agree with those reported by Barrientos-Velázquez et al. (2016) [82], who found that montmorillonite clay showed a higher adsorption capacity for AFB_1_ (0.52 mol/kg) than that of beidellite (0.25 mol/kg). They attributed this to the presence of a greater amount of charge on the octahedral sheets in comparison to the tetrahedral sheets, which increased the interactions with the AFB_1_ molecules. Wang et al. (2020) [83] reported that the binding sites for AFB_1_ in smectites lie within the interlayer spaces and that the interactions occur with the carbonyl group of AFB_1_.

Dvorák (1989) [84] reported that the retention rate of AF molecules in a culture medium of *Aspergillus parasiticus* NRRL 2999 was higher for smectite samples (66%) than that for zeolite samples (60%). Similarly, Nuryono et al. (2012) [85] observed that the adsorption rate of AFB_1_ in contaminated corn samples was 99% for smectite and 96% for zeolite. These results are consistent with those observed in the present study. However, Savari et al. (2013) [86] observed that clinoptilolite presented a higher AFB_1_ adsorption rate in contaminated rice samples compared with smectite. Additionally, Tomasevic-Canovic et al. (2001) [87] and Bočarov-Stančić et al. (2018) [88] did not observe any differences in the AFB_1_ adsorption capacities of montmorillonite and clinoptilolite. Further, a change in the pH of the medium induces a loss of Al^3+^ ions from the tetrahedral sheets of clinoptilolite, leading to the formation of new binding active sites [51]. Another characteristic of this mineral is its porous structure, which allows for dipolar interactions with water and polar molecules [89]. Other researchers have reported that the concentrations of Ca^2+^ and K^+^ ions have a positive effect on the CEC values. Specifically, an increase in the concentration of these ions increases the adsorption of AFB_1_. In this study, elemental composition measurements showed that C-3 clay had the highest Ca^2+^, K^+^, Al^3+^, and Fe^3+^ ion contents, and C-4 clay had the highest Ca^2+^, K^+^, and Al^3+^ ion contents but the lowest AFB_1_ adsorption capacity. In contrast, Ayo et al. (2018) [90] reported that Si^4+^, Al^3+^, and Fe^3+^ ions had a negative influence on the CEC and decreased the adsorption of AFB_1_.

### 3.4. Trace Inorganic Nutrient Adsorption

Trace inorganic nutrients are important for physiological functions (growth, development, and health) in broilers [91]. The deficiency of these essential nutrients may result in reduced synthesis of enzymes and proteins such as collagen, a main component of feathers and beaks, and keratin, a structural protein of the extracellular matrix and connective tissue [92,93]. Side effects include severe anemia, loss of pigmentation in feathers, and bone malformation [94]. Trace inorganic nutrients participate also as cofactors of enzymes, acting as antioxidant molecules to control the production of free radicals [95]. Aluminosilicates can form complexes with vitamins and trace minerals [30], and depends on the type, temperature, and pH of the medium [96]. 

We observed that while all the clays adsorbed trace minerals, C-3 exhibited the highest adsorption capacity (Fe > Se > Zn > Mn), followed by C-4, C-2, and C-1 clay. The CEC in smectites is the result of substituting Al^3+^ for Si^4+^ in the tetrahedral sheet, and Mg^2+^ for Al^3+^ in the octahedral sheet, producing a negative layer charge. Similarly, zeolites substitute Al^3+^ for Si^4+^ in the tetrahedral sheet. These results agree with the elemental composition of C-3 clay and C-4 clay, which presented a higher content of alumina, in contrast with C-1 clay and C-2 clay. This suggests that the adsorption by the active centers of the adsorbents depends specifically on the content of the alumina and silica ions [97].

Other studies have reported that Fe and Zn adsorption by montmorillonite depends on the pH [96,98] and particle size [66]. In contrast, Van Groeningen et al. (2020) [99] observed that the adsorption of Mn by aluminosilicates is not pH-dependent.

## 4. Conclusions

This study evaluated the chemical and mineral compositions of three smectites and one zeolite and how these affect the adsorption of AFB_1_ and trace inorganic nutrients. The results obtained showed that the mechanism of AFB_1_ adsorption of clays is through an interaction with their functional groups (Si-O, Al-O, Si-O-Si, and Si-O-Al), generating changes in the crystallinity structure such as porous size, substitution of ions, and lattice strain. This effect is attributable to the octahedral and tetrahedral structures and the alumina and silica content. Although clays adsorb AFB_1_, they also capture trace inorganic nutrients, affecting animal nutrition and productivity. Of the four clays evaluated, C-1 (montmorillonite) showed the highest AFB_1_ adsorption and the lowest adsorption of all trace inorganic nutrients (except Fe), suggesting that a competition between these molecules was not apparent. Further studies should be conducted to determine the in vivo adsorption capacity of smectites for AFB_1_ and to investigate the adverse effects of binding trace inorganic nutrients included in poultry diets.

## 5. Materials and Methods

### 5.1. Reagents

AFB_1_ from *Aspergillus flavus* (catalog no. A6636), pepsin from porcine gastric mucosa (catalog no. P7000), pancreatin from porcine pancreas (catalog no. P7545), ammonium acetate (catalog no. A7262), ammonium chloride (catalog no. 213330), ammonium hydroxide (catalog no. 221228), ethylenediaminetetraacetic acid (EDTA) (catalog no. EDS-60004), and Eriochrome Black T (EBT) (catalog no. 858390) were purchased from Sigma-Aldrich (Burlington, MA, USA). Acetonitrile (catalog no. 75-05-8), methanol (catalog no. 67-56-1), and HPLC-grade water (catalog no. 7732-18-5) were purchased from Fisher Scientific (Waltham, MA, USA). The clays used and trace inorganic nutrients (iron sulfate, manganese sulfate, zinc sulfate, and sodium sulfate) were provided by MNA de México (Juárez, México).

### 5.2. Characterization of Clays

The four clays, denoted as C-1, C-2, C-3 clays (smectites), and C-4 clay (zeolite), were characterized using FTIR spectroscopy, XRD analysis, XRF measurements, and SEM. The functional groups of clays and their interactions with AFB_1_ were determined using FTIR spectroscopy. The clays were placed onto a diamond ATR crystal, and the spectra were recorded using an FTIR spectrometer (Perkin Elmer; Waltham, MA, USA) based on 16 accumulated scans at a resolution of 4 cm^−1^ between 3800 and 400 cm^−1^. The data were analyzed using Spectrum 10 software (Perkin Elmer).

The mineralogical compositions of the clays were determined using an XRD analyzer (BTX SN 231, Olympus Corporation; Tokyo, Japan). Specifically, 15 mg of the finely ground (150 µm) test sample was loaded into the vibration holder of the XRD analyzer for scanning. The XRD patterns were analyzed for the peaks corresponding to the minerals in the clay sample, and its quantitative composition was estimated using the Rietveld method. Chemical analysis was performed using the XRF method in accordance with standard procedures. The morphologies and structures of the clays were characterized using ultrahigh-resolution field-emission SEM (Hitachi SU8020; Schaumburg, IL, USA).

### 5.3. Determination of Cation Exchange Capacity (CEC)

The CEC value was determined using an ammonium acetate solution to extract the exchangeable cations present in the clay samples [100]. A 0.5 g sample of the clay to be tested was mixed with 100 mL of a 1 N ammonium acetate salt solution and shaken for 5 min. Next, the sample was filtered, and 5 mL of the solution was mixed with 10 mL of ammonium buffer solution. Finally, the sample was titrated with a standard EDTA solution using 10 drops of EBT. The endpoint was indicated by a color change from red to blue. The CEC was calculated using the following equations:(1)$$meq\;exch\;divalent\;cations=\left(mL\;EDTA\right)\left(molarity\;of\;EDTA\right)\left(2\right)\left(100\frac{g}{wt}of\;sample\right)$$
*meq total exch. bases* = (*meq*
*exch divalent cations*) (1.05)(2)
(3)$$CEC=\frac{meq\;total\;exch\;bases}{\%\;base\;saturation/100}$$

### 5.4. Measurement of AFB_1_ Adsorption Capacity by Ultra Performance Liquid Chromatography (UPLC) Using an In Vitro Poultry Gastrointestinal Model

An in vitro gastrointestinal model for poultry was used to determine the AFB_1_ adsorption capacities of the various clay samples [101]. Three compartments of the poultry model were simulated: (1) crop with a pH of 5.2, under constant agitation and incubation at 40 °C for 30 min; (2) proventriculus with a pH of 2.5 and 3000 U of pepsin, under constant agitation and incubation at 40 °C for 45 min; and (3) small intestine compartment with a pH of 6.6 and 6.84 mg of pancreatin, under constant agitation and incubation at 40 °C for 2 h.

For the in vitro adsorption experiments, a stock solution of AFB_1_ (5000 µg/2 mL) was prepared with acetonitrile and diluted with distilled water to a concentration of 4 µg/mL. Next, 40 mg of the clay to be analyzed was weighed and mixed with the AFB_1_ solution (4 µg/mL, 0.3 mL) and water (5.7 mL) to reach a final AFB_1_ concentration of 200 ng/mL. The AFB_1_: clay mass ratio was calculated based on specifications of commercial clays (2 kg/ton) and according to the Official Mexican Standard NOM-188-SSA1-2002, USDA, and FDA (maximum level of AFB_1_ of 20 ppb in poultry feed), respectively. The pH of the compartments was adjusted to 2.5, 5.2, or 6.6. We also used two sets of control samples: (1) water at a pH of 2.5, 5.2, or 6.6 with AFB_1_ but no clay; and (2) water at a pH of 2.5, 5.2, or 6.6 with clay but no AFB_1_. The AFB_1_ solution and clay samples were mixed and incubated under the conditions corresponding to the in vitro poultry gastrointestinal model described above to quantify the AFB_1_ concentration in the supernatant. After each incubation period, the sample was centrifuged at 2000 rpm for 10 min, and the supernatant was filtered through a 0.2 µm syringe filter (Whatman^®^ UNIFLO^®^; Burlington, MA, USA). Next, the supernatant was evaporated in a nitrogen stream at 50 °C, and the obtained residue was dissolved in 500 µL of the mobile phase (6.4:1.8:1.8 water/methanol/acetonitrile). Subsequently, the obtained solution was analyzed using a Waters ACQUITY UPLC^®^ H-Class Bio System (Milford, MA, USA) coupled to a fluorescence detector and an ACQUITY UPLC BEH C18 phase reverse column (2.1 mm × 50 mm, 1.7 µm). Specifically, 10 µL of the solution were injected and eluted with the mobile phase at a flow rate of 0.5 mL/min; the excitation and emission wavelengths were 365 and 429 nm, respectively. AFB_1_ was identified using a retention time of 4 min. For reference, we used a pure AFB_1_ standard solution with a concentration of 200 ng/mL.

Each sample was measured in quadruplicate, and the concentration of AFB_1_ remaining was calculated using a curve calibration with *R*^2^ = 0.996. The amount of AFB_1_ adsorbed (percentage) was calculated from the difference between the peak area of the control (AFB_1_) and that of the test clay using the following equation:*R* = [(1 − *A*1)/*A*0] (100)(4)
where *A*0 corresponds to the peak area of the control (AFB_1_) and *A*1 to the peak area of the adsorbent (i.e., test clay).

### 5.5. Measurement of Trace Inorganic Nutrient Adsorption Rate

The adsorption percentages of the trace inorganic nutrients (Fe, Mn, Zn, and Se) were measured using a standard solution containing the nutritional requirements for poultry (Table 1) [62]. The inorganic sources of the trace inorganic nutrients were dissolved in 500 mL of distilled water, and 40 mg of the adsorbent was weighed and mixed with 6 mL of this trace inorganic nutrients solution. The sample was incubated under conditions corresponding to the in vitro poultry gastrointestinal model, after which it was centrifuged at 2000 rpm for 20 min. The trace inorganic nutrient concentrations in the supernatant were determined using inductively coupled plasma optical emission spectroscopy (Perkin Elmer OPTIMA 2000 DV; Waltham, MA, USA).

### 5.6. Statistical Analysis

The experimental data were analyzed with the analytical software Statistics 9 (Tallahassee, FL, USA) using a completely randomized design. The data were subjected to one-way analysis of variance (ANOVA), and the means were compared using Tukey’s test at *p* ≤ 0.05.

## Figures and Tables

**Figure 1 toxins-15-00519-f001:**
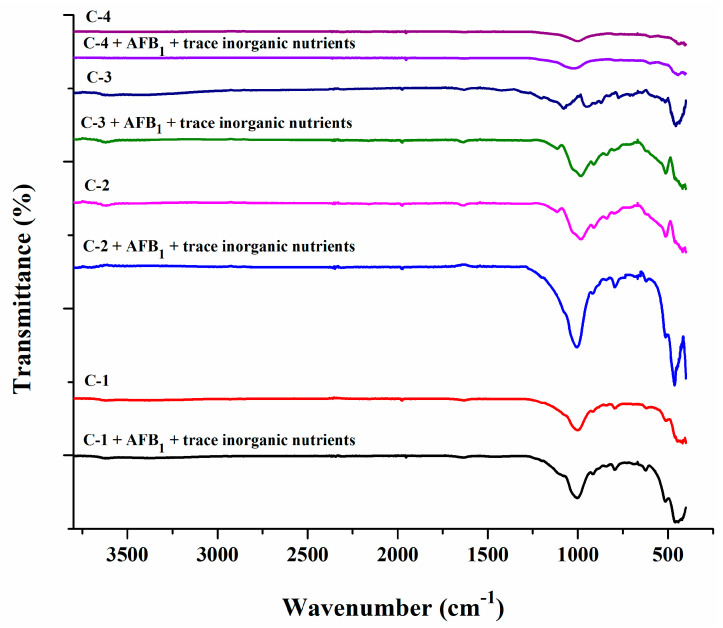
FTIR spectra of the clay minerals (C-1, C-2, C-3, and C-4) before and after adsorption of molecules (AFB_1_ and trace inorganic nutrients).

**Figure 2 toxins-15-00519-f002:**
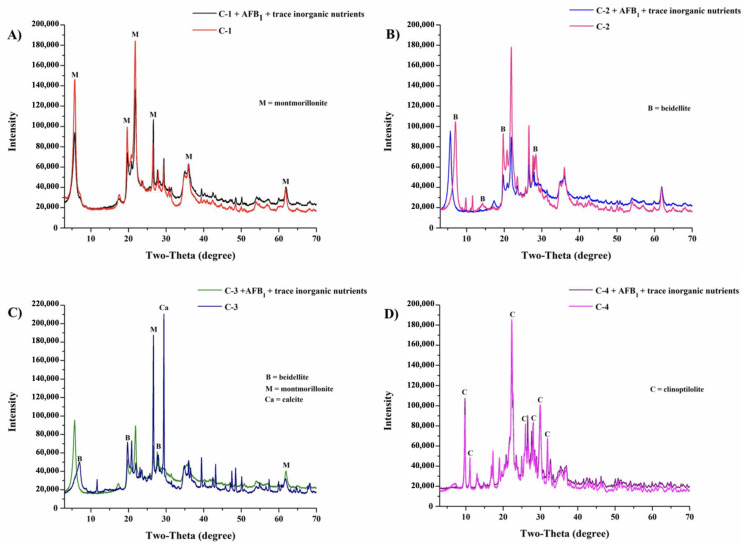
XRD patterns of the clay minerals before and after adsorption of molecules (AFB_1_ and trace inorganic nutrients). (**A**) C-1 clay (Ca-montmorillonite), (**B**) C-2 clay (beidellite), (**C**) C-3 clay (beidellite-Ca-montmorillonite), and (**D**) C-4 clay (clinoptilolite).

**Figure 3 toxins-15-00519-f003:**
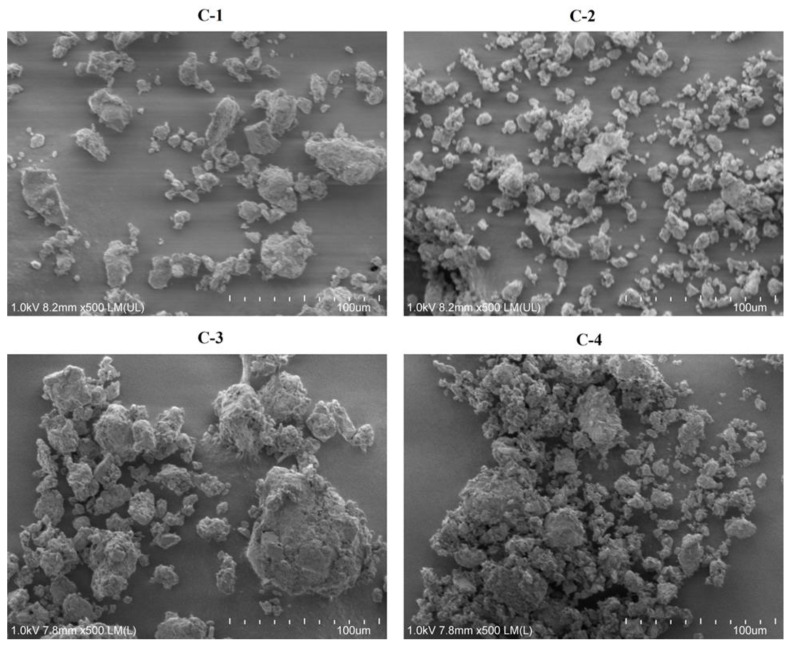
Scanning electron microscope images of C-1 clay (Ca-montmorillonite), C-2 clay (beidellite), C-3 clay (beidellite-Ca-montmorillonite), and C-4 clay (clinoptilolite).

**Figure 4 toxins-15-00519-f004:**
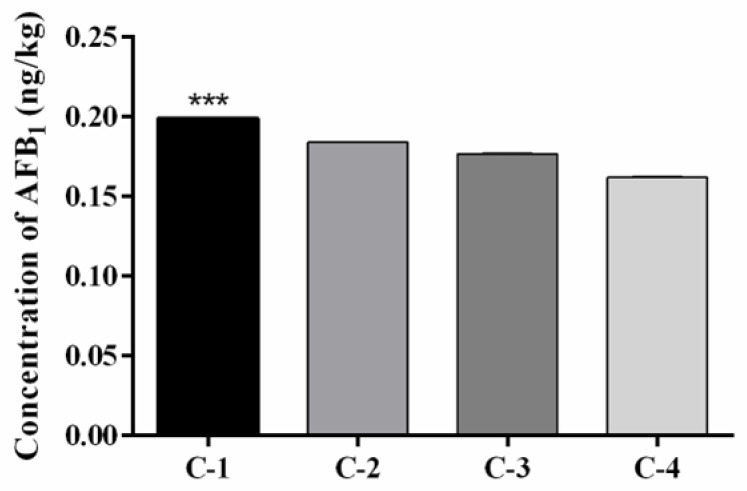
Concentration of AFB_1_ adsorbed in the small intestine compartment for C-1 clay (Ca-montmorillonite), C-2 clay (beidellite), C-3 clay (beidellite-Ca-montmorillonite), and C-4 clay (clinoptilolite). *** *p* = 0.001.

**Figure 5 toxins-15-00519-f005:**
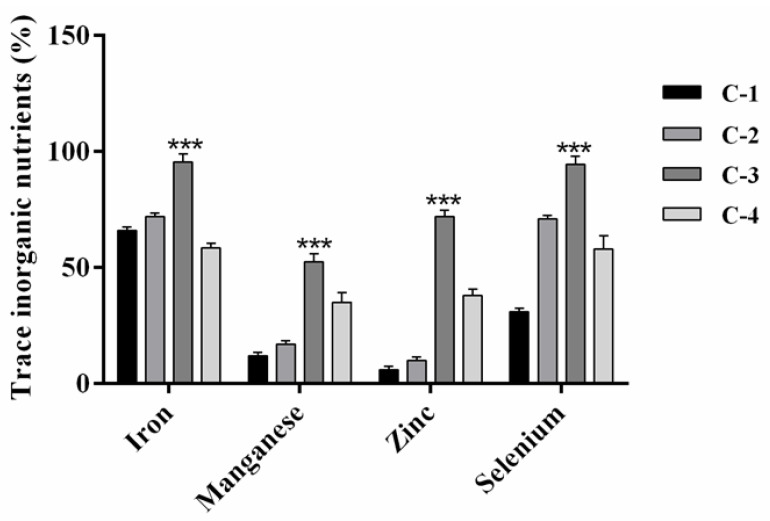
Percentage of adsorption of trace inorganic nutrients in the small intestine compartment for C-1 clay (Ca-montmorillonite), C-2 clay (beidellite), C-3 clay (beidellite-Ca-montmorillonite), and C-4 clay (clinoptilolite) in the in vitro gastrointestinal model for poultry. *** *p* = 0.001.

**Table 1 toxins-15-00519-t001:** Elemental composition of clay minerals (%).

Clay Samples	Fe_2_O_3_	TiO_2_	CaO	K_2_O	SiO_2_	Al_2_O_3_	MgO	Na_2_O	SO_3_
C-1	2.56	0.19	2.23	0.48	69.37	9.14	5.02	0.25	0.10
C-2	2.31	0.20	1.40	1.21	72.71	11.07	2.64	1.91	1.05
C-3	5.47	0.63	9.48	2.23	50.22	13.97	3.77	2.28	1.01
C-4	2.25	0.28	2.96	4.99	65.79	13.33	1.32	0.46	0.20

**Table 2 toxins-15-00519-t002:** Mineralogical composition of clay minerals.

Clay Samples	Mineral	Percentage (%)	Chemical Formula
C-1	Montmorillonite	61.0	Ca_0.2_(Al,Mg)_2_Si_4_O_10_(OH)_2_·4H_2_O
	Cristobalite	21.0	SiO_2_
	Beidellite	10.0	Na_0.3_Al_2_(Si,Al)_4_O_10_(OH)_2_·2H_2_O
	Orthoclase	2.5	KAlSi_3_O_8_
	Quartz	5.5	SiO_2_
C-2	Beidellite	55.0	Na_0.3_Al_2_(Si,Al)_4_O_10_(OH)_2_·2H_2_O
	Cristobalite	24.0	SiO_2_
	Orthoclase	7.0	KAlSi_3_O_8_
	Quartz	6.0	SiO_2_
	Albite	3.5	NaAlSi_3_O_8_
	Clinoptilolite	3.0	(Na_0.52_K_2.44_Ca_1.48_)(Al_6.59_Si_29.41_O_72_)(H_2_O)_28.64_
	Gypsum	1.5	CaSO_4_·2H_2_O
C-3	Beidellite	36.5	Na_0.3_Al_2_(Si,Al)_4_O_10_(OH)_2_·2H_2_O
	Montmorillonite	15.0	Ca_0.2_(Al,Mg)_2_Si_4_O_10_(OH)_2_·4H_2_O
	Illite	14.0	K(Al_4_Si_2_O_9_)(OH)_3_)
	Calcite	12.0	CaCO_3_
	Albite	10.0	NaAlSi_3_O_8_
	Quartz	7.0	SiO_2_
	Sanidine	4.0	KAlSi_3_O_8_
	Gypsum	1.5	CaSO_4_·2H_2_O
C-4	Clinoptilolite	80.0	(Na_0.52_K_2.44_Ca_1.48_)(Al_6.59_Si_29.41_O_72_)(H_2_O)_28.64_
	Sanidine	8.0	KAlSi_3_O_8_
	Montmorillonite	5.0	Na_0.3_(Al,Mg)_2_Si_4_O_10_(OH)_2_·4H_2_O
	Cristobalite	4.0	SiO_2_
	Illite	2.0	K(Al_4_Si_2_O_9_)(OH)_3_)
	Quartz	1.0	SiO_2_

**Table 3 toxins-15-00519-t003:** Mineralogical composition of clay minerals.

Clay Samples	CEC (cmol(+)/kg)
C-1	57.44
C-2	54.72
C-3	60.18
C-4	38.29

**Table 4 toxins-15-00519-t004:** Concentration of AFB_1_ adsorbed by different clay minerals in the small intestine compartment.

Clay Samples	AFB_1_ Adsorbed (ng/kg)	AFB_1_ Adsorbed (%)
C-1	0.199 ± 0.00012	99.5
C-2	0.184 ± 0.00013	92.02
C-3	0.176 ± 0.00016	88.40
C-4	0.161 ± 0.0004	80.97

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
