# Peer review of "Aluminosilicates as a Double-Edged Sword: Adsorption of Aflatoxin B1 and Sequestration of Essential Trace Minerals in an In Vitro Gastrointestinal Poultry Model"

_toxins, 2023, doi:10.3390/toxins15090519_

Round 1

Reviewer 1 Report

The manuscript investigated composition and structural characteristics of three smectites and one zeolite, and evaluated them as adsorbents of AFB1 using in-vitro poultry gastrointestinal model. The study designed well and obtained some interesting information. However, some comments need to be addressed before the manuscript can be accepted:

1. Line 40: “AB1-N7-guanine adduct” should be “AFB1-N7-guanine adduct”

2. Lines 50-54: “These molecules induce fluorescence which can be detected… chemiluminescence immunoassays, and immunosensors”. The fluorescence cannot be detected by the methods of HPLC and mass spectroscopy, rewrite the sentence, please.

3. Line 63: “high-temperature” method belongs to physical method. Rewrite the sentence please.

4. In the Introduction section (lines 65-66), clay minerals as one of the means preventing the toxic effects of Afs in animal nutrition was introduced. To arouse readers' interest, it is necessary to briefly introduce other detoxifying means used in animal nutrition in this paragraph, except for the detoxification mean of aflatxoins introduced herein, such as animal function regulator (curcumin), microbial methods (probiotics, non-toxigenic Aspergillus), and allow me to suggest the following publication to be cited herein, please:

Wang, Y.; Liu, F.; Zhou, X.; Liu, M.; Zang, H.; Liu, X.; Shan, A.; Feng, X. Alleviation of Oral Exposure to Aflatoxin B1-Induced Renal Dysfunction, Oxidative Stress, and Cell Apoptosis in Mice Kidney by Curcumin. Antioxidants 2022, 11, 1082. https://doi.org/10.3390/antiox11061082

5. Figure 1: “FTIR spectra of adsorption molecules (trace inorganic nutrients) are missing in the figure, add them please. The range of horizontal coordinate values is too narrow to embody the vibrational bands corresponding to H-O-H bending of water between 3,600 cm-1 and 1,600 cm-1 and bands at 3,717 cm-1 and 1,629 cm-1 related with vibration of OH groups of water.

6. Table 1: the value (72.7) of SiO2 percentage in the sample C-2 only contains one significant figure after the decimal separator. Provide the value containing two significant figures after the decimal separator consistent with that of other data in the table, please.

7. Lines 127-130: the paragraph should be rewritten according to the data presented in Table 1. “alumina (50–73%) and silica (9–14%), should be SiO2 (50–73%) and Al2O3 (9–14%). “magnesium”, “iron”, “calcium” and “potassium” in the sentence “The other minerals included magnesium in C-1 clay (5.02%), iron (5.47%) and calcium (9.48%) in C-3 clay, and potassium (4.99%) in C-4 clay” should be changed to the names of their oxides.

8. Table 2: “Various components account for 96% of the sample C-4. What are the other 4% components?

9. Figure 2: the information of “trace inorganic nutrients” is missing the figure. Add it in the figure, please.

10. Table 3: Provide the values containing two significant figures after the decimal separator in the table as elucidated in lines 164-165, please.

11. Lines 361-365, the presentation of the three equations should be improved. For example: there are no spaces between words, the letter “x” is confused with the multiplication sign “×”, Annotations for equations are missing.

12. Line 379: delete the word “used”, please.

13. Line 402: the letter “x” in the equation should be changed to the multiplication sign “×”.

Minor editing of English language required.

Author Response

Dear Reviewer

Thank you for your observations, we improved our title and article, and all the suggestions and comments were considered. Please advise if this is acceptable.

1. Line 40: “AB1-N7-guanine adduct” should be “AFB1-N7- guanine adduct”.

            In line 42 we do the change.

2. Lines 50-54: “These molecules induce fluorescence which can be detected... chemiluminescence immunoassays, and immunosensors”. The fluorescence cannot be detected by the methods of HPLC and mass spectroscopy, rewrite the sentence, please.

            We improve the paragraph in line 53-57.

3. Line 63: “high-temperature” method belongs to physical method. Rewrite the sentence please.

            In line 66 we rewrite the sentence.

4. In the Introduction section (lines 65-66), clay minerals as one of the means preventing the toxic effects of Afs in animal nutrition was introduced. To arouse readers' interest, it is necessary to briefly introduce other detoxifying means used in animal nutrition in this paragraph, except for the detoxification mean of aflatxoins introduced herein, such as animal function regulator (curcumin), microbial methods (probiotics, non-toxigenic Aspergillus), and allow me to suggest the following publication to be cited herein, please:

Wang, Y.; Liu, F.; Zhou, X.; Liu, M.; Zang, H.; Liu, X.; Shan, A.; Feng, X. Alleviation of Oral Exposure to Aflatoxin B1- Induced Renal Dysfunction, Oxidative Stress, and Cell Apoptosis in Mice Kidney by Curcumin. Antioxidants 2022, 11, 1082. https://doi.org/10.3390/antiox11061082

            We add the reference in line 68-78.

5. Figure 1: “FTIR spectra of adsorption molecules (trace inorganic nutrients) are missing in the figure, add them please. The range of horizontal coordinate values is too narrow to embody the vibrational bands corresponding to H- O-H bending of water between 3,600 cm-1 and 1,600 cm-1 and bands at 3,717 cm-1 and 1,629 cm-1 related with vibration of OH groups of water.

            The range was increased in Figure 1; 3,800 to 400 cm-1.

6. Table 1: the value (72.7) of SiO2 percentage in the sample C-2 only contains one significant figure after the decimal separator. Provide the value containing two significant figures after the decimal separator consistent with that of other data in the table, please.

            We added the decimal values and was corrected to 72.71.

7. Lines 127-130: the paragraph should be rewritten according to the data presented in Table 1. “alumina (50– 73%) and silica (9–14%), should be SiO2 (50–73%) and Al2O3 (9–14%). “magnesium”, “iron”, “calcium” and “potassium” in the sentence “The other minerals included magnesium in C-1 clay (5.02%), iron (5.47%) and calcium (9.48%) in C-3 clay, and potassium (4.99%) in C-4 clay” should be changed to the names of their oxides.

            We rewrote the paragraph in line 145-148.

8. Table 2: “Various components account for 96% of the sample C-4. What are the other 4% components?

            There was an error in Table 2 in the Sanidine; the correct percent is 8.

9. Figure 2: the information of “trace inorganic nutrients” is missing the figure. Add it in the figure, please.

            We arrangement Figure 2; added the trace inorganic nutrients.

10. Table 3: Provide the values containing two significant figures after the decimal separator in the table as elucidated in lines 164-165, please.

            We added the decimal values.

11. Lines 361-365, the presentation of the three equations should be improved. For example: there are no spaces between words, the letter “x” is confused with the multiplication sign “×”, Annotations for equations are missing.

            We adjust the equations.

12. Line 379: delete the word “used”, please.

            We deleted the word “used”.

13. Line 402: the letter “x” in the equation should be changed to the multiplication sign “×”.

            We change the equation.

Reviewer 2 Report

I regret that I am not able to critically assess portions of the paper – specifically the physical chemistry components.  Regarding the aflatoxin binding, the work is OK, but not novel.  The binding has been described in many papers for many years, and I am not clear what is unique about the present manuscript.  Much of this paper seems to be simply describing the physical characteristics of the clays, with the aflatoxin aspect handled as a minor afterthought.  

The work appears to have been done competently and the writing is, overall fairly clear.  

Line 7:  “commonly”?  I know that there has been a lot of literature on use of clay adsorbents, but my understanding is that in the US none were registered / labeled for aflatoxin mitigation.  They are used in the poultry industry, but not officially for aflatoxin mitigation.  If I’m wrong, provide documentation.

Line 65-70:  Again, many papers have been published on this, and I understand that some livestock feeders may use this with this intent, but to my knowledge no product is approved specifically for this use.  If I’m wrong, provide documentation.

Line 114:  You describe a range “between 3,600 cm-1 and 1,600 cm-1 “, but the figure only starts at 2500, and I don’t see anything before ca. 1,250.

Table 4 and Figure 4:  These two items seem to be describing the same thing – Table 4 is the amount bound to the clay and Figure 4 is the amount the clay did not bind.  Is this correct?  If so, this is probably redundant.  

Author Response

Dear Reviewer,

Thank you for your observations, we improved our title and article, and all the suggestions and comments were considered. Please advise if this is acceptable.

Comments and Suggestions for Authors:

I regret that I am not able to critically assess portions of the paper – specifically the physical chemistry components. Regarding the aflatoxin binding, the work is OK, but not novel. The binding has been described in many papers for many years, and I am not clear what is unique about the present manuscript. Much of this paper seems to be simply describing the physical characteristics of the clays, with the aflatoxin aspect handled as a minor afterthought. The work appears to have been done competently and the writing is, overall fairly clear.

Dear reviewer, considering your observation that our work is not novel, we decided to change the title of the paper to: “Aluminosilicates as a double-edged sword: adsorption of aflatoxin B1 but sequestration of essential trace minerals in in-vitro gastrointestinal poultry model”. Please advise if this is acceptable.

We included the following paragraph in the introduction: “The binding capacity of clays is specific and depends on their structure, even if they belong to the same family [30]. However, they also sequester other molecules such as trace inorganic nutrients (Zn, Mn, Fe, and Se) that are important for physiological functions (growth, development, and health) in broilers [31]. The deficiency of these essential nutrients may result in reduce synthesis of enzymes and proteins such as collagen, a main component of feathers, beaks, and keratin, a structural protein of the extracellular matrix and connective tissue [32,33]. Side effects include severe anemia, loss of pigmentation in feathers and bone malformation [34]”.

Line 7: “commonly”? I know that there has been a lot of literature on use of clay adsorbents, but my understanding is that in the US none were registered / labeled for aflatoxin mitigation. They are used in the poultry industry, but not officially for aflatoxin mitigation. If I’m wrong, provide documentation.

            Considering your observation, we did a search in the literature and just found a register (GRAS Notice No. AGRN 29) of a clinoptilolite as an anticaking in diets for cattle, swine, goats, sheep, poultry, cats, and dogs.

Line 65-70: Again, many papers have been published on this, and I understand that some livestock feeders may use this with this intent, but to my knowledge no product is approved specifically for this use. If I’m wrong, provide documentation.

            Considering your observation, we do a search in the literature and just found a register (GRAS Notice No. AGRN 29) of a clinoptilolite as an anticaking in diets for cattle, swine, goats, sheep, poultry, cats, and dogs.

Line 114: You describe a range “between 3,600 cm-1 and 1,600 cm-1 “, but the figure only starts at 2500, and I don’t see anything before ca. 1,250.

            We change the figure.

Table 4 and Figure 4: These two items seem to be describing the same thing – Table 4 is the amount bound to the clay and Figure 4 is the amount the clay did not bind. Is this correct? If so, this is probably redundant.

            The results of the Table and Figure 4 are the same.

Reviewer 3 Report

The present manuscript entitled "Aluminosilicates as adsorbents of aflatoxin B1 and trace miner-2 als in in-vitro gastrointestinal poultry model” presents data in the efficacy of four types of clays in feed to adsorb aflatoxins in poultry. Also, present data in adverse effect of the clays by adsorbing trace mineral nutrients essential in poultry diets. The manuscript is well written, and the methods used are sound and seems well performed. This manuscript is acceptable for publication in Toxins after minor suggested edits.

The main issue with the manuscript is that the authors present results of the adsorption of aflatoxins by different types of clays, but to a rather low concentration of aflatoxin (20 ng/g), which is the allowed concentration for human and animal consumption. At this level the results indicate a good efficiency of the clays to adsorb aflatoxins, with higher efficiency from some clay types. However, concentration higher than this level, which will be the ones considered to be harmful and cause problems of poisoning by aflatoxins, were not evaluated. So, what will happen if the aflatoxin concentrations in the feed are higher? Will these clays have a point of saturation and not be able to adsorb any excess of aflatoxins and therefore still be problems of poisoning? Or will aflatoxins still be adsorbed if the clays are saturated? Aflatoxin at 20 ng/g is not supposed to be a problem, at least poultry are more sensitive to aflatoxins and the 20 ng/g levels represent a problem. The authors should address this on the discussion section. Also, will be advisable to test higher concentrations, which will be the ones that will cause poisoning by aflatoxin. How much clay will need to be added so it will be safe for both nutritionally and free of aflatoxins? It will be good to add a few sentences discussing this issue.

Some specific edits that need to be fixed:

P1/L5       Remove the comma (,) after Aflatoxins. It should read “Aflatoxins can cause …”

P1/L8       It should read “to prevent the absorption of the carcinogens …” Aflatoxins to be harmful need to be absorbed into the digestive system of animals and humans. The clays adsorb the aflatoxins to avoid the absorption.

P1/L28      Should include tree nuts (which include pistachio, almond and walnuts). They are very important crops affected by aflatoxins.

P1/L31      Drop ‘and humid’. It should read “usually grow in warm conditions …” Even though humidity is needed for the fungi growth, excess humidity in the tropical areas are not conducive to Aspergillus growth.

P2/L46        Should include US, so to read “The US Food and Drug….”

P2/L60-64   The authors mention several methods to prevent contamination and exposure to aflatoxins, which are not very reliable, but fail to mention the use of atoxigenic strains of A. flavus as biocontrol, which is the only proven method to prevent contamination and is being used commercially. It will be useful to discuss its use to strengthen the manuscript.

Author Response

Dear Reviewer,

Thank you for your observations, we improved our title and article, and all the suggestions and comments were considered. Please advise if this is acceptable.

Comments and Suggestions for Authors:

The present manuscript entitled "Aluminosilicates as adsorbents of aflatoxin B1 and trace minerals in in-vitro gastrointestinal poultry model” presents data in the efficacy of four types of clays in feed to adsorb aflatoxins in poultry. Also, present data in adverse effect of the clays by adsorbing trace mineral nutrients essential in poultry diets. The manuscript is well written, and the methods used are sound and seems well performed. This manuscript is acceptable for publication in Toxins after minor suggested edits.

The main issue with the manuscript is that the authors present results of the adsorption of aflatoxins by different types of clays, but to a rather low concentration of aflatoxin (20 ng/g), which is the allowed concentration for human and animal consumption. At this level the results indicate a good efficiency of the clays to adsorb aflatoxins, with higher efficiency from some clay types. However, concentration higher than this level, which will be the ones considered to be harmful and cause problems of poisoning by aflatoxins, were not evaluated. So, what will happen if the aflatoxin concentrations in the feed are higher? Will these clays have a point of saturation and not be able to adsorb any excess of aflatoxins and therefore still be problems of poisoning? Or will aflatoxins still be adsorbed if the clays are saturated? Aflatoxin at 20 ng/g is not supposed to be a problem, at least poultry are more sensitive to aflatoxins and the 20 ng/g levels represent a problem. The authors should address this on the discussion section. Also, will be advisable to test higher concentrations, which will be the ones that will cause poisoning by aflatoxin. How much clay will need to be added so it will be safe for both nutritionally and free of aflatoxins? It will be good to add a few sentences discussing this issue.

            We use this concentration of AFs because is the allowed value for human and animal consumption. In the case this concentration increases, it is necessary to increase the level of clay inclusion since it can become saturated and increasing the clay could cause a nutritional imbalance.

1. P1/L5 Remove the comma (,) after Aflatoxins. It should read “Aflatoxins can cause ...”

         In line 6 the comma was removed.

P1/L8 It should read “to prevent the absorption of the carcinogens ...” Aflatoxins to be harmful need to be absorbed into the digestive system of animals and humans. The clays adsorb the aflatoxins to avoid the absorption.

            In line 8 and 9 the paragraph was rearranged.

P1/L28 Should include tree nuts (which include pistachio, almond and walnuts). They are very important crops affected by aflatoxins.

            We include in line 30 the pistachio, almond, and walnuts.

P1/L31 Drop ‘and humid’. It should read “usually grow in warm conditions ...” Even though humidity is needed for the fungi growth, excess humidity in the tropical areas are not conducive to Aspergillus growth.

            In line 33 and 34 we change the sentence.

P2/L46 Should include US, so to read “The US Food and Drug....”

            In line 48 we added US.

P2/L60-64 The authors mention several methods to prevent contamination and exposure to aflatoxins, which are not very reliable, but fail to mention the use of atoxigenic strains of A. flavus as biocontrol, which is the only proven method to prevent contamination and is being used commercially. It will be useful to discuss its use to strengthen the manuscript.

            This paragraph was included in the introduction: “Another strategy is the utilization of atoxigenic A. flavus strain (non-aflatoxin) to limit aflatoxin contamination [28]. The atoxigenic strain decreased the production of AFs by competition for nutrients and regulation of aflatoxin biosynthesis [29]”.

Round 2

Reviewer 1 Report

All my comments have been addressed in the revised manuscript, and I consider that it is suitable for publication in the journal of "Toxins".

Minor editing of English language required.